# Signaling Pathway Mediating Myeloma Cell Growth and Survival

**DOI:** 10.3390/cancers13020216

**Published:** 2021-01-08

**Authors:** Teru Hideshima, Kenneth C. Anderson

**Affiliations:** Jerome Lipper Multiple Myeloma Center, Department of Medical Oncology, Dana-Farber Cancer Institute, Harvard Medical School, 450 Brookline Avenue, Boston, MA 02446, USA

**Keywords:** multiple myeloma, bone marrow microenvironment, cytokine, cell adhesion, signal transduction

## Abstract

**Simple Summary:**

The bone marrow (BM) microenvironment plays a crucial role in pathogenesis of multiple myeloma (MM), and delineation of the intracellular signaling pathways activated in the BM microenvironment in MM cells is essential to develop novel therapeutic strategies to improve patient outcome.

**Abstract:**

The multiple myeloma (MM) bone marrow (BM) microenvironment consists of different types of accessory cells. Both soluble factors (i.e., cytokines) secreted from these cells and adhesion of MM cells to these cells play crucial roles in activation of intracellular signaling pathways mediating MM cell growth, survival, migration, and drug resistance. Importantly, there is crosstalk between the signaling pathways, increasing the complexity of signal transduction networks in MM cells in the BM microenvironment, highlighting the requirement for combination treatment strategies to blocking multiple signaling pathways.

## 1. Introduction

MM cells interact with different types of cells in the BM microenvironment. These cells include BM stromal cells, osteoclasts, osteoblasts, vascular endothelial cells, and immune effector cells. Although genetic abnormalities have been shown to play a crucial role in disease evolution and progression in hematologic malignancies, the BM microenvironment also plays a major contribution in MM pathogenesis by activating intracellular signaling pathways that promote growth, survival, and migration of MM cells, as well as resistance to conventional chemotherapeutic and novel agents [1]. Recent advances of single cell analyses of DNA, RNA, and protein also reveal diversity of MM cell intracellular signaling. Targeting proteins mediating these signaling pathways therefore offers a potential therapeutic strategy to inhibit MM cell growth and overcome drug resistance in the BM milieu. In this review, we will discuss representative major signaling pathways in MM cells which are relevant in the context of the BM microenvironment.

## 2. Cellular Components Interacting with MM Cells in the BM Microenvironment

The BM microenvironment is composed of various cell types including BM stromal cells (BMSCs), osteoclasts, osteoblasts, dendritic cells, and vascular endothelial cells. There are established signal networks between MM cells and those cellular components that not only stimulate, but also suppress proliferation of MM cells, and vice versa. For example, MM cells suppress the differentiation and proliferation of osteoblasts, and together with the induction of osteoclast differentiation and hyperfunction, are factors in the formation of osteolytic lesions in MM [1].

## 3. Mediators for Activation of MM Signaling Pathways

The interactions between MM cells and other cellular components in the BM microenvironment activate intracellular signaling pathways in MM cells via three routes. The first mechanism is by soluble factors including cytokines and chemokines secreted by accessory cells. For example, IL-6, IL-17, oncostatin-M (OSM), leukemia inhibitory factor (LIF), insulin-like growth factor-1 (IGF1), vascular endothelial growth factor (VEGF), fibroblast growth factor (FGF), CXCL12 (also known as stromal-cell-derived factor-1α; SDF1α), and tumor necrosis factor-α (TNF-α) are secreted from BMSCs [1,2]. On the other hand, osteoclasts produce IL-6 and VEGF, which are growth factors for MM cells [3]. Cyclophilin-A is secreted from vascular endothelial cells and plays an important role in the proliferation of MM cells and their homing to the BM milieu [4]. MM cells also produce dickkopf-related protein 1 (DKK1) and macrophage inflammatory protein 1α (MIP1α), which inhibit the Wnt pathway mediating bone formation. To define the role of these soluble factors in progression of MM, plasma levels of IL-6, b-FGF, hepatocyte growth factor (HGF), VEGF and transforming growth factor-β_1_ (TGF-β_1_), as well as soluble receptors for IL-6 (sIL-6R) and VEGF (sVEGF-R2) were measured. Plasma levels of HGF, b-FGF, IL-6 and sIL-6R in MM patients were higher than those in the control group [5]. Moreover, the immunologic and pro-angiogenic soluble factors and their downstream signaling pathways also impact disease progression from monoclonal gammopathy of undetermined significance (MGUS) to MM. A recent study assessed 45 immunologic and pro-angiogenic markers in sera from 241 MM case patients, 441 participants with non-progressing MGUS, and 258 MGUS-free control participants. Importantly, the top six markers, including epidermal growth factor (EGF), HGF, angiopoietin (Ang)-2, SDF-1α, monocyte chemotactic protein (ΜCP), and bone morphogenetic protein (BMP)-9, were significantly associated with progression of MGUS to MM [6]. HGS can also potentiate impact of IL-6 on MM cell proliferation and migration [7].

The second route is physical interaction of MM cells with other cellular components via cell adhesion molecules. For example, intercellular adhesion molecule 1 (ICAM-1) and vascular cell adhesion molecule 1 (VCAM-1) on MM cells bind lymphocyte function-associated antigen-1 (LFA-1) [8] and very late antigen (VLA-4) [9,10] on BMSCs, respectively. This intercellular adhesion activates proliferative and/or anti-apoptotic signals in MM cells. On the other hand, NF-κB pathway can be activated in BMSCs by MM cell adhesion, resulting in upregulated transcription and secretion of cytokines (i.e., IL-6) and adhesion molecules, which further enhance MM cell proliferation and activate anti-apoptotic signaling pathways [1,11].

The third route is via exosomes which are small membrane vesicles secreted by various cell types including dendritic cells, B-cells, T-cells, mast cells, epithelial cells, and tumor cells. Exosomes contain cell-specific, bioactive molecules, and exert their functions by transferring their cargo to their target cells either by endocytosis or by direct fusion with the cell membrane. MM BM-mesenchymal stromal cell (MSC)-derived exosomes, defined as CD63+/CD81+, promote MM tumor growth via downregulation of a tumor suppressor miR-15a; in contrast, normal BM-MSC exosomes inhibit growth of MM cells [12]. Importantly, BMSC-derived exosomes activate several survival pathways in MM cells including c-Jun N-terminal kinase, p38, p53, and Akt, which also play an important role mediating drug resistance in MM [13]. Exosomes from BMSCs and MM cells enhance MM progression by the induction of drug resistance, angiogenesis, and/or host immune suppression [14]. Taken together, soluble factors from accessory cells and MM cells, as well as interactions/adhesion of MM to accessory cells, forms a complex network regulating MM pathogenesis and response to therapy (Figure 1). Although exosomes play an important role on MM cell pathogenesis in the BM microenvironment, they cannot be classified as classical soluble factors and are not further discussed.

## 4. Overview of Signaling Pathways Activated in MM Cells in the BM Microenvironment

The BM microenvironment consists of different cell types that produce and secrete numerous soluble factors and physically interact with MM cells. Importantly, soluble activating factors are further upregulated by MM cell adhesion with these cells [1,2,11,15,16,17,18,19,20,21,22,23,24,25,26,27,28,29,30]. These cytokines activated signaling pathways include: extracellular signal related kinase (ERK, also known as p42/44 mitogen activated protein kinase); Janus kinase (JAK)2/signal transducers and activators of transcription (STAT)3; phosphoinositide 3-kinases (PI3K)/Akt (also known as protein kinase B); nuclear factor κ-B (NF-κB); transforming growth factor (TGF)-β/Smad and Wnt/β-catenin (Table 1).

Downstream sequelae include: cytoplasmic sequestration of many transcription factors (i.e., FKHRL-1); upregulation of cell cycle regulatory proteins (i.e., D-type cyclins) and anti-apoptotic proteins (i.e., Bcl-2, Bcl-xL, Mcl-1, caspase inhibitors); increased activity of telomerase [24]. Importantly, all these events promote MM cell growth and survival.

Among the cells promoting MM cell proliferation and drug resistance, BMSCs are implicated in both cytokine- and cell adhesion-mediated signal transduction. Many growth factors secreted by MM cells and BMSCs also stimulate osteoclastogenesis (IL-6, IL-1, VEGF, SDF-1α, MIP-1α) and angiogenesis (VEGF). Moreover, genetic lesions in MM may modulate the ability of MM cells to interact with their BM milieu. For example, MM cells with t (14; 16) translocation overexpress the transcription factor MAF [31,32], which not only transactivates the cyclin D2 promoter, but also upregulates β7-integrin expression, thereby enhancing MM cell adhesion to BMSCs [33]. Bone marrow endothelial cells (BMECs) secrete SDF-1α, which mediates the initial homing of MM cells to the BM stromal compartment through CXCR4. Adhesion between MM cells and BMECs then upregulates many cytokines with angiogenic activity. MM cells also constitutively produce, due to oncogene activation and/or genetic mutations, factors such as VEGF, basic fibroblast growth factor (bFGF), and matrix metalloproteinases (MMP)s that stimulate BM angiogenesis [34]. Conversely, BMECs secrete growth factors including VEGF, IL-6, and IGF-1, which promote MM cell growth in the BM milieu [35]. The BMECs in these new MM-associated vessels further support MM cells through cytokines and direct adhesive interactions.

The interaction of MM cells and osteoclasts (OCLs) mediated via both cytokines and physical interaction is also an important factor not only in bone resorption, but also in MM cell growth. Specifically, osteoclasts produce osteopontin (OPN) and IL-6, and adhesion of MM cells to OCLs further increases IL-6 secretion from OCLs. Moreover, combination of IL-6 and OPN markedly enhances MM cell growth and survival [36].

## 5. Soluble Factor-Mediated Signaling Pathways in MM

### 5.1. p42/p44 Mitogen-Activated Protein Kinase (MAPK; ERK1/2) Signaling Pathway

p42/p44 MAPKs (ERK1/2) are a widely conserved and well validated family of serine/threonine protein kinases mediating cellular programs such as proliferation, differentiation, and motility. Although mutation of either ERK, or MEK has not yet been reported, mutations of upstream kinases K-ras and N-ras which induce constitutive ERK activation are common in MM [37,38,39]. Moreover, activation of ERK in MM cells may be associated with progression of the disease. A previous study demonstrated that the mean tumor burden and median survival for patients with mutations of N-ras was indistinguishable from patients with no *RAS* mutations; on the other hand, patients with *K-ras* mutations had a significantly higher mean bone marrow tumor burden at diagnosis than patients with no ras mutations [38]. In contrast, Martin et al. reported the absence of mutations within either codon 12 of *K-ras* or codon 61 of *N-ras* in MGUS or MM, suggesting that Ras mutations may not play a significant role in the pathogenesis of MM in Spanish population [40]. The ERK1/2 signaling pathway can be further activated in response to a diverse range of extracellular stimuli including mitogens, cytokines, and chemokines. Upon stimulation, a sequential three-part protein kinase cascade is initiated consisting of MAP kinase kinase kinase (MAPKKK, Raf), MAP kinase kinase (MAPKK, MEK), and MAP kinase, (ERK). In MM cells, ERK1/2 is constitutively activated (phosphorylated), which is further enhanced by many cytokines in the BM microenvironment including IL-6, VEGF, BAFF, CXCL12, and Wnt [1,2].

Recurrent chromosomal translocations occur in approximately 40% of MM patients. These rearrangements are due to aberrant class-switching, resulting in the linkage of the immunoglobulin promoter/enhancer to the *Cyclin D1*, *Cyclin D3*, *MMSET* and *FGFR3*, *c-MAF* or *MAF-B* genes. Translocation t (14; 16) leading to elevated expression of *MAF* occurs in approximately 10% of cases. High levels of *MAF* are also observed in the cases with t (4; 14) translocation, associated with overexpression of *MMSET* and *FGFR3*. Importantly, MEK-ERK pathway also regulates *MAF* transcription to promote MM cell proliferation and drug resistance [41]. Most recently, activating mutations in ERK pathway promoting resistance to proteasome inhibitor by increasing proteasome capacity has been demonstrated [42], suggesting that activation of MEK/ERK pathway in the BM microenvironment may confer clinical proteasome inhibitor resistance.

Another important feature of MEK/ERK signaling is its crosstalk with other major signaling pathways including JAK2/STAT3 and PI3K-Akt. For example, ERK positively regulates serine (Ser727) phosphorylation of STAT3, which mediates transcriptional activity of STAT3 [43]. However, Serine 727 phosphorylation may negatively modulate STAT3 tyrosine phosphorylation, which is required for dimer formation, nuclear translocation, and DNA binding activity of STAT3. Therefore, activation of ERK2 is able to inhibit STAT3 function [44,45]. We have also shown that PI3K inhibitor downregulates MEK-ERK phosphorylation in MM cells, suggesting that PI3K and/or its downstream molecules positively regulate the MEK-ERK pathway [46].

As described above, ERK signaling mediates MM cell proliferation, and targeting this pathway therefore represents an attractive therapeutic option. We have already shown that knocking down ERK by antisense oligonucleotide results in significant MM cell growth inhibition in vitro [47]. Numerous efforts have been made to target MEK-ERK pathway using small molecule MEK inhibitors. AZD6244 (selumetinib) is also a potent ATP-non-competitive allosteric MEK inhibitor that has demonstrated high activity in both in vitro and in vivo tumor xenograft models. Specifically, it targeted both MM cells and osteoclasts derived from BM-derived macrophages [48]. Although promising anti-MM activities of AZD6244 were shown in preclinical studies, it showed only minimal activity in patients with MM [49]. Current ongoing clinical evaluation of MEK inhibitors, alone or in combination with other agents, in MM include RO5126766 (ClinicalTrials.gov Identifier: NCT02407509, Phase I), BIBW2992 (ClinicalTrials.gov Identifier: NCT02465060, Phase II), and binimetinib (ClinicalTrials.gov Identifier: NCT02834364, Phase II), as well as Trametinib + Akt inhibitor GSK2141795 (ClinicalTrials.gov Identifier: NCT01989598, Phase II).

Whole-genome sequencing data have recently revealed that a subset of patients possesses an activating mutation (*V600E*) in the BRAF kinase. The *BRAF V600E* mutation triggers constitutive activation of Ras-Raf-MEK-ERK signaling pathway, stimulating cellular growth, differentiation and survival [50]. However, it is still unclear whether constitutively activated MEK-ERK pathway in the cells with BRAF V600E is further enhanced by soluble activating factors in the BM microenvironment. A previous study showed that BRAF V600E was detected in 2.8% of patients with symptomatic myeloma and 1.8% of all patients with monoclonal plasma cell disorders, respectively [51]. Another study also showed that BRAF V600E was detected in 5.3% of the patients with MM [52]. Importantly, patients with BRAF V600E showed significantly higher incidence of extramedullary disease and a shorter overall survival [51]. Sorafenib is an orally available compound which predominantly inhibits Raf kinase and VEGF receptor 2. In a phase II clinical trial (SWOG S0434) of sorafenib as monotherapy in relapsed and refractory MM (RRMM), it did not exhibit clinical activity [53]. Further research should focus on combination therapy of sorafenib with standard treatments in selected patients with RR MM [53]. For example, sorafenib upregulates Akt phosphorylation, and its use in combination with rapamycin to inhibit mTOR [54] showed synergistic growth inhibitory effects.

### 5.2. JAK2-STAT3 Signaling Pathway

As described above, many cytokines and chemokines in the BM microenvironment trigger activation of signaling cascades. Importantly, JAK2-STAT3 pathway in MM cells can be activated only by gp130 receptor cytokine family members, such as IL-6, leukemia inhibitory factor (LIF) and oncostatin-M (OSM). Among these cytokines, IL-6 is the most potent activator of JAK2-STAT3 pathway in the BM microenvironment [1]. Multiple reports support an autocrine IL-6–mediated growth mechanism in MM, since some MM cells and derived cell lines both produce and respond to IL-6 in vitro [55]. Autocrine IL-6 production is associated with a highly malignant phenotype, high proliferative index, and resistance to drug-induced apoptosis [56]. However, most IL-6 in the BM microenvironment is secreted by BMSCs, and its transcription and secretion is regulated by NF-κB [16]. Moreover, IL-6 secretion from BMSCs is further augmented both by soluble factors which can activate NF-κB (i.e., TNFα) and MM cell adhesion [16,57,58,59,60,61].

JAK2 is a JAK family member non-receptor tyrosine kinase (RTK) which is highly expressed in MM cells. It is tightly associated with IL-6R β-subunit (gp130). Upon IL-6 binding to IL-6R α-subunit (gp80), JAK2 is phosphorylated at Tyr1034/1035 and subsequently induces phosphorylation of tyrosine residues (Tyr759) of gp130, followed by interaction and phosphorylation (Tyr705) of transcription factor STAT3 [62,63,64,65], which in turn triggers dimerization and nuclear translocation of STAT3 [66]. This JAK2-STAT3 interaction can be inhibited by negative feedback loop via suppressor of cytokine signaling 3 (SOCS3), which is a transcriptional target of STAT3. As described above, phosphorylation of serine residue (Ser727) involved in STAT3 transcriptional activity is positively mediated by ERK.

The most important biologic sequelae of JAK2-STAT3 pathway in MM cells is to maintain survival by regulating expression of downstream anti-apoptotic Bcl2 family proteins, including Bcl-XL, Mcl-1 and survivin [67,68]. Recent studies have shown that IL6 activates STAT3 which regulates transcription of phosphatase of regenerating liver-3 (PRL-3), and that upregulation of PRL-3 increases MM cell viability and re-phosphorylates STAT3 through direct interaction and deactivation of SHP2. Furthermore, loss of PRL-3 abolishes nuclear localization of STAT3, thereby decreasing its localization on the promoter of target genes such as c-Myc and Mcl-1, Bcl2 and Bcl-xL [69].

Monoclonal antibody-based immunotherapies are emerging therapeutic options in MM. Specifically, anti-CD38 monoclonal antibody treatments including daratumumab are effective therapies for both newly diagnosed and relapsed MM. We have shown that BMSC culture supernatant and IL-6 downregulate CD38 expression via activation of STAT3, resulting in reduced antibody-dependent cellular cytotoxicity (ADCC) by daratumumab. These results suggest that daratumumab-induced ADCC may be attenuated in the BM microenvironment. Conversely, a JAK inhibitor ruxolitinib blocks STAT3 phosphorylation and upregulates CD38 expression, thereby augmenting daratumumab-induced ADCC against MM cell lines in vitro [70].

Based on preclinical and clinical studies, JAK2-STAT3 pathway is a promising therapeutic target in MM. Indeed, many efforts have focused on developing small molecule STAT3 inhibitors. However, to date, no clinical grade small molecule selective STAT3 inhibitor has been evaluated, at least in part due to difficulty generating inhibitors against transcription factors. Therefore, using inhibitors against upstream molecule JAK2 may be practical. Early results from the oral treatment regimen of a JAK2 inhibitor ruxolitinib, corticosteroids (methylprednisolone), and lenalidomide for patients with RRMM are encouraging in terms of safety and efficacy, and further studies may provide support for this promising new therapeutic option [71]. The other approach to inhibit STAT3 activity in MM is to utilize histone deacetylase inhibitor (HDACi). Panobinostat is a non-selective HDAC inhibitor, which was approved by FDA in 2015 to treat MM and which has progressed after treatment with at least two prior standard therapies. In preclinical studies, we have shown that non-selective HDACi including Panobinostat significantly reduced both constitutive and IL-6-induced p-STAT3 in MM cell lines [72].

### 5.3. PI3K-Akt Signaling Pathway

The PI3K family is divided into four different classes: Class I, Class II, Class III, and Class IV. We will specifically focus on Class I PI3Ks, since they have been extensively studied and validated in many cancers including MM. PI3K is composed of a heterodimer of p85 (regulatory domain) and p110 (catalytic domain), and p110 is further divided into several isoforms. For example, there are five variants of the p85 regulatory subunit called p85α, p55α, p50α, p85β, and p55γ. There are also three variants of the p110 catalytic subunit called the p110α, β, or δ catalytic subunits. Activated PI3K subsequently activates the downstream molecule phosphoinositide dependent kinase 1 (PDK1). PDK1 is a downstream kinase of PI3K that plays a crucial role in activation of Akt (also known as protein kinase B) [73,74]. Importantly, phosphatase and tensin homologue deleted on chromosome ten (PTEN) is a major negative regulator of the PI3K-Akt signaling pathway [75]. Therefore, PTEN gene abnormalities (inactivating) could induce constitutive Akt activation. Indeed, PTEN gene abnormalities with high Akt activity are known in MM cell lines (i.e., OPM2 cells); however, they are not common events in patient MM cells [76,77].

In MM, PI3K-Akt signaling is one of the most complex signaling pathways since it is activated by many cytokines including IL-6, IGF-1, VEGF, CXCL12, and BAFF [1,11]; it has numerous downstream targets mediating diverse biological activities. Furthermore, there is crosstalk between PI3K-Akt and other signaling pathways including ERK and NF-κB. It is constitutively activated and can be further enhanced by soluble factors in the BM microenvironment. In particular, IL-6 and IGF1 strongly stimulate this pathway and mediate cell growth, cell cycle entry, cell migration, and cell survival [78]. Many molecules are identified as downstream of Akt. For example, protein kinase glycogen synthase kinase (GSK), Bcl-2 family member Bad, inhibitor of apoptosis protein (IAP) family member survivin [79], MDM2, and forkhead transcription factor (FKHR) have been reported in MM cells. Therefore, PI3K-Akt pathway is a potent mediator of survival, anti-apoptosis, and resistance in MM cells.

One of the unique features of this signaling pathway is control of autophagy. The autophagic pathway and the proteasome pathway regulate the degradation of intracellular proteins in MM. Recent studies show that Akt regulates downstream mTOR (mammalian target of rapamycin) and autophagy via the tuberous sclerosis 2 (TSC) 1/2 complex [80,81]. Therefore, inhibition of Akt or mTORC1 triggers autophagy. Moreover, there is a crosstalk (inhibition and activation) between the Ras-EK-ERK and PI3K-Akt pathways [82,83,84]. More specifically, components of the Ras-MEK-ERK pathway (Ras, Raf, ERK, RSK) positively regulate PI3K-mTORC1 pathway [82]. We have also shown that Akt inhibitor IL-6-Hydroxymethyl-chiro-inositol 2-(R)-2-O-methyl-3-O-octadecylcarbonate induces MM cell death associated with downregulation of the NF-κB activity, suggesting crosstalk between PI3K-Akt and NF-κB signaling pathways [85]. It also modulates cell cycle and proliferation both directly via activity on the CDK inhibitors p21^WAF1/Cip1^ and p27^Kip1^, as well as indirectly by affecting the levels of p53 and cyclinD1. Moreover, Akt mediates cell survival directly by inhibiting proapoptotic Bad and indirectly by modulating major regulators of cell death, such as p53, NF-κB, and telomerase [78,86,87,88,89,90,91]. Since PI3k-Akt signaling is a promising therapeutic target in cancers, a number of inhibitors against PI3K, Akt and/or mTOR have been developed [92], which show significant anti-tumor (including MM) activity, either alone or in combination with other therapeutic agents or radiotherapy.

In MM, we have shown that non-specific Akt inhibitor perifosine induces significant cytotoxicity in MM cells which are resistant to conventional therapeutic agents. In addition, perifosine synergistically enhances MEK inhibitor- or proteasome inhibitor bortezomib-induced cytotoxicity [30]. A Phase I, multicenter, single-arm study to assess the safety and determine the maximum-tolerated dose of perifosine-lenalidomide-dexamethasone in relapsed and relapsed/refractory MM has been conducted. Among 30 evaluable patients, 73% achieved a minimal response or better, including 50% with a partial response or better. Median progression-free survival was 10.8 months, and median overall survival 30.6 months. Response was associated with inhibition of phospho-Akt in pharmacodynamic studies. Perifosine-lenalidomide-dexamethasone was well tolerated [93]. MK2206 is an allosteric Akt inhibitor and also shows anti-MM activity in the preclinical setting [94]. Specifically, MK2206 induces cytotoxicity and inhibits proliferation in all MM cell lines tested, albeit with significant heterogeneity dependent on basal pAkt levels. MK2206 was able to inhibit proliferation of MM cells, even when cultured with marrow stromal cells or tumor promoting cytokines [95]. Moreover, MK2206 enhances the cytocidal effects of bufalin in MM cells [94]. A phase II study is evaluating MK2206 in patients with relapsed or refractory lymphoma of any histology excluding Burkitt lymphoma or lymphoblastic lymphoma was carried out (ClinicalTrials.gov Identifier: NCT01258998) [96]; however, it has not yet been tested in clinical trials in MM. We have shown that TAS-117, a highly selective non-ATP competitive small molecule inhibitor of AKT, triggers both apoptosis and autophagy, associated with induction of endoplasmic reticulum stress response. TAS-117 enhances bortezomib-induced cytotoxicity in vivo in a murine model of human MM, with prolonged host survival [97]. GSK2141795 (Uprosertib) is an ATP-competitive and orally bioavailable Akt inhibitor. A Phase II clinical trial of an MEK 1/2 Inhibitor Trametinib and GSK2141795 is treating patients with RRMM (ClinicalTrials.gov Identifier: NCT01989598). This combination treatment was also evaluated in patients with acute myeloid leukemia, and showed no clinical activity in patients with *RAS*-mutated AML [98].

### 5.4. NF-κB Signaling Pathway

NF-κB, like STAT3, is a transcription factor that is activated by humoral factors such as TNFα, IL-1β, CD40, and BAFF in MM cells in the BM microenvironment. NF-κB is a member of Rel family proteins including RelA (p65), RelB, c-Rel, NFκB1 (p50) and NFκB2 (p52), which can form homodimers or heterodimers. However, its DNA binding activity and/or transcriptional activity varies depending on the combination. For example, homodimers of p50 have high DNA binding ability, but low transcriptional activity. Activation of this pathway is also regulated in multiple steps like other signaling pathways. Classically, NF-κB signaling is divided into two pathways, the canonical and non-canonical pathways in which NFκB1 (p50)-RelA (p65) and NFκB2 (p52)-RelB, respectively, act as NF-κB transcription factors. Activation of NF-κB by the canonical and non-canonical pathways is induced by different mechanisms: the canonical pathway is regulated by the trimer of IκB kinasse (IKK) α, IKKβ, and IKKγ; the non-canonical pathway is regulated by the dimer of IKKα. Furthermore, TNFα and IL-1β predominantly activate the canonical pathway, whereas CD40 and BAFF activate the non-canonical pathway. Importantly, both NF-κB pathways are constitutively activated in MM cells in the BM microenvironment [99]. There is a crosstalk between these pathways, and when one pathway is blocked, the other can demonstrate compensatory activation [99]. In the canonical pathway, a heterodimer composed of p50 and p65 subunits is constitutively present in the cytosol and nucleus. In the cytosol, NF-κB is inactivated by its association with IκB family inhibitors, such as IκBα [100]. IκBα therefore has a crucial role in regulating canonical NF-κB activation. For example, various growth and/or anti-apoptosis promoting cytokines, including tumor necrosis factor (TNF)α [61,101,102] and IGF-1 [85], trigger phosphorylation of IκB proteins by upstream IκB kinases (IKKs), followed by its proteasomal degradation. These events allow for nuclear translocation of NF-κB, where it binds to specific DNA sequences in the promoters of target DNA, thereby facilitating transcription. However, the details of target gene expression specific to each pathway are not totally understood.

The precise biologic and functional roles of NF-κB signaling pathway, canonical vs non-canonical, in MM has not been fully characterized. However, it regulates numerous cytokines, chemokines, cell adhesion molecules, and cell cycle regulators, as well as anti-apoptotic proteins and drug-resistance mediators in cancers, including MM [103,104]. Importantly, genetic abnormalities associated with NF-κB activation have also been reported in MM [105,106], confirming that non-canonical NF-κB signaling plays an important role in MM pathogenesis. Specifically, the non-canonical NF-κB pathway is constitutively activated in MM cells with inactivation of TRAF3 [106], suggesting that the non-canonical pathway represents a novel therapeutic target, and that inhibition of only the canonical pathway may be insufficient to fully block NF-κB activity.

Since NF-κB signaling pathway is also regulated in multiple steps, NF-κB activity can be inhibited at different levels. Importantly, the majority of currently available small molecule NF-κB inhibitors target IKKβ and IκBα that are mediators of the canonical pathway. These agents directly inhibit IKKβ activity, blocking phosphorylation of both p65 and IκBα. Downregulation of phosphorylation in p65 and IκBα decreases p65 transcriptional activity and proteasomal degradation of IκBα, respectively. We have shown that small molecule IKKβ inhibitors PS-1145 and MLN120B block MM cell growth in the context of BMSCs, associated with downregulation of IL-6 secretion from BMSCs [102,107]. However, MLN120B showed only modest inhibition of MM cell growth in a clinically relevant SCID-hu mouse model (human bone chip in SCID mouse) [107]. These results suggest the requirement for dual inhibitor or combination clinical treatments to target both canonical and non-canonical NF-κB pathways. Since degradation of IκBα is proteasome dependent, proteasome inhibitors can stabilize IκBα protein level, thereby inhibiting canonical NF-κB activity by preventing p50–p65 nuclear translocation. Since conversion of p100 to p52 (NFκB2) is proteasome dependent [108], proteasome inhibitors are also able to block non-canonical NF-κB pathway.

### 5.5. Wnt β-Catenin Signaling

Wnts are produced by a variety of cell types including epithelial cells in the BM microenvironment, and have been implicated in hematopoietic stem cell maintenance [109] and early B cell development [110]. Wnts comprise a family of secreted proteins and interact with receptors, consisting of a Frizzled (Fz) family member alone or complexed with LDL receptor-related proteins (LRP5/6). Wnt signaling mediates various developmental processes and can lead to malignant transformation. There are three Wnt signaling pathways: the canonical Wnt pathway, the non-canonical planar cell polarity pathway, and the non-canonical Wnt-calcium pathway. These pathways are activated by the binding of a Wnt-protein ligand to a Frizzled family receptor, which transmits the biological signal to the Dishevelled protein inside the cell. The canonical Wnt pathway leads to regulation of gene transcription, and is negatively regulated in part by the SPATS1 gene. The non-canonical planar cell polarity pathway regulates the cytoskeleton that is responsible for the shape of the cell. The non-canonical Wnt/calcium pathway regulates calcium inside the cell [111]. Intracellularly, Wnt signaling pathway blocks phosphorylation and degradation of multifunctional β-catenin by proteasomes, thereby leading to its accumulation in the cytoplasm [112].

The biologic significance of Wnt-β-catenin signaling in MM has been studied [111,113]. In MM, the canonical Wnt signaling pathway can be activated by Wnt-3a, associated with accumulation of β-catenin. Wnt-3a treatment also triggers significant morphological changes in MM cells, accompanied by rearrangement of the actin cytoskeleton [114]. A previous study demonstrated that MM cells overexpress β-catenin [115], including its N-terminally unphosphorylated form, consistent with activated β-catenin-T cell factor (TCF)-mediated transcription. Moreover, increased nuclear localization of β-catenin associated with proliferation can be mediated by stimulation of Wnt signaling via Wnt-3a or constitutively active mutations of β-catenin [116]. Importantly, discordant results regarding the role of Wnt3a-mediated MM cell growth stimulation have been reported: Wnt3a activates canonical pathway in the majority of MM cell lines and patient MM cells; however, it has no effect on MM cell growth in vitro and in vivo in the SCID-hu model [114].

We have shown that F115–584, which disrupts the interaction of the transcriptionally active β-catenin-TCF protein complex, blocks expression of Wnt target genes and induces significant cytotoxicity in both MM cell lines and patient MM cells, without toxicity in normal plasma cells. In xenograft models of human MM, PKF115-584 inhibits MM tumor growth and prolongs host survival [117]. Interestingly, Quing et al. demonstrated that Wnt-mediated migration is associated with the Wnt-RhoA pathway, but does not require signaling through β-catenin [118]. Importantly, MM cells in BM-biopsy specimens contained detectable dickkopf 1 (DKK1), a negative regulator of Wnt signaling cascade and target of the β-catenin-TCF pathway. Moreover, elevated DKK1 levels in BM plasma and peripheral blood from patients with MM correlated with the DKK1 gene-expression patterns associated with focal bone lesions [119]. However, a previous study showed that MM cells do not demonstrate inhibition of canonical Wnt signaling in the human BM microenvironment [120].

We have also shown that extracellular cyclophilin-A (eCyPA) secreted by BM endothelial cells (BMECs) promotes the colonization and proliferation of MM cells via binding to its receptor CD147 on MM cells in preclinical setting. Expression and secretion of eCyPA by BMECs was enhanced by a Wnt-β-catenin transcriptional coactivator BCL9. Importantly, eCyPA levels are higher in BM serum than in peripheral blood in individuals with MM, and eCyPA-CD147 blockade suppressed MM colonization and tumor growth in vivo in murine models. These findings suggest that eCyPA-CD147 signaling promotes homing of MM cells to the BM, and provide the rationale for further validating this axis as a therapeutic target for MM [4]. In addition to effect in MM cells, Wnt signaling also tightly controls the balance between osteoblasts and osteoclasts by direct and indirect mechanisms, respectively [121].

## 6. Cell Adhesion-Induced Signaling Pathways

Physical interaction of MM cell with other cellular components in the BM microenvironment triggers activation of signaling pathways in both MM cells and accessory cells. This cell-cell adhesion is mediated via matching of specific adhesion molecules on each cell surface; VLA-4 on MM cells and VCAM-1 on BMSC. This cell adhesion not only induces signals mediating proliferation of MM cells via ERK (p42/44 MAPK), but also upregulates production of cytokines and the expression of adhesion molecules in BMSCs via activation of NF-kB. Specifically, we have previously shown that MM cell adhesion to BMSCs induces NF-κB activation in BMSCs, which enhances transcription and secretion of IL-6 [16,102]. Since IL-6 is a major growth and survival factor in MM cells [122,123], and adherence of MM cells to BMSCs confers resistance to conventional drug-induced apoptosis, specific blockade of NF-κB signaling in BMSCs represents a novel therapeutic strategy in MM (Figure 2). Of note, IL-6 also enhances angiogenesis via VEGF stimulation and modulates osteoclast differentiation [124].

One of the prominent features of cell adhesion-mediated signaling in MM pathogenesis is cell-adhesion mediated drug resistance (CAM-DR). This phenomenon was initially reported based upon MM cell adhesion to fibronectin (FN) via VLA-4 and VLA-5 mediating MM cell survival and anti-apoptosis [19]. Subsequently, adhesion of MM cells to FN was shown to result in G_1_ arrest associated with increased p27^*kip*1^ protein levels, as well as inhibition of cyclin A and E associated kinase activity. Treatment of cells with p27^*Kip*1^ antisense oligonucleotides re-sensitized MM cells to cytotoxic drugs. These results suggest that disruption of β1 integrin mediated FN adhesion may represent a potential target for the potentiation of drug-induced apoptosis in MM [125]. Moreover, another study reported that CAM-DR is associated with activation of canonical NF-κB (p65–p50) in MM cells. In this study, NF-*κ*B binding activity was significantly increased when cells adhered to FN compared to cells in suspension [126]. We have confirmed that MM cell adhesion to BMSCs activates both canonical and non-canonical NF-κB pathways in MM cells [99]. Taken together, these studies demonstrate that MM cell adhesion to BMSCs triggers NF-κB activation in both MM cells and BMSCs.

Since expression of the majority of adhesion molecules including VLA-4 is regulated by NF-κB, inhibition of NF-κB may represent a novel strategy to overcome CAM-DR. As described in Section 5.4, bortezomib is able to block both canonical and non-canonical NF-κB pathways by inhibiting proteasomal degradation of IκBα and p100 conversion to p52 (NFκB2), respectively. Indeed, an earlier study has shown that VLA-4 plays a critical role in CAM-DR of MM cells, and that combination of bortezomib with conventional anti-MM drugs may be effective in overcoming CAM-DR [127].

CXCL12 (also known as SDF1α) is indispensable for homeostatic processes such as lymphopoiesis and embryogenesis. CXCL12 is constitutively produced by BMSCs [128]. We have shown that CXCL12 induces phosphorylation of ERK and Akt, as well as NF-κB, thereby promoting tumor cell growth, survival, and migration. It also protects MM cells from dexamethasone-induced apoptosis [23]. A recent study has also shown that CXCL12 is a relevant target to attenuate CAM-DR in MM [129]. Alterations in chromatin modifications, such as histone methylation, mediate chemotherapy resistance in several cancer types. Kikuchi et al. demonstrated that pharmacological and genetic inhibition of the IGF1R-PI3K-AKT pathway reverses CAM-DR by promoting enhancer of zeste homolog 2 (EZH2) dephosphorylation and H3K27 hypermethylation both in vitro and in refractory murine MM models, suggesting an epigenetic mechanism underlying CAM-DR [130].

## 7. Conclusions

In the BM microenvironment, intracellular signaling cascades in MM cells are continuously activated by soluble factors and contact to other cellular components, which promotes survival, proliferation, and drug resistance of MM cells. Signal transduction in myeloma cells is very complicated in the BM microenvironment, and multiple signal transduction systems are simultaneously activated by multiple soluble factors. For example, one soluble factor may activate a plurality of signal transduction pathways at the same time, but a plurality of soluble factors may cooperate to induce more potent activation of the same signaling pathway. It should also be noted that the existence of other factors that could suppress myeloma cells growth cannot be ruled out. Therefore, it should be carefully evaluated whether the findings obtained in the preclinical studies actually occur in the BM environment of MM patients. Therefore, combination treatments to inhibit these soluble factors and/or their mediators of intracellular signaling cascades may represent promising novel therapeutic strategies in MM.

## Figures and Tables

**Figure 1 cancers-13-00216-f001:**
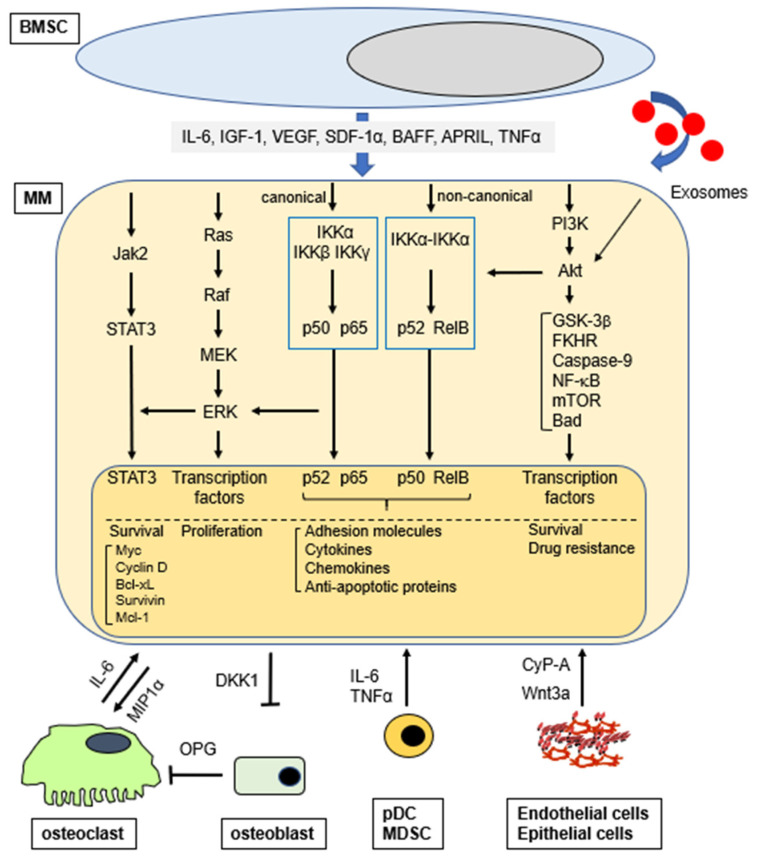
Activation of signaling pathways (selected) in MM cells in the BM microenvironment.

**Figure 2 cancers-13-00216-f002:**
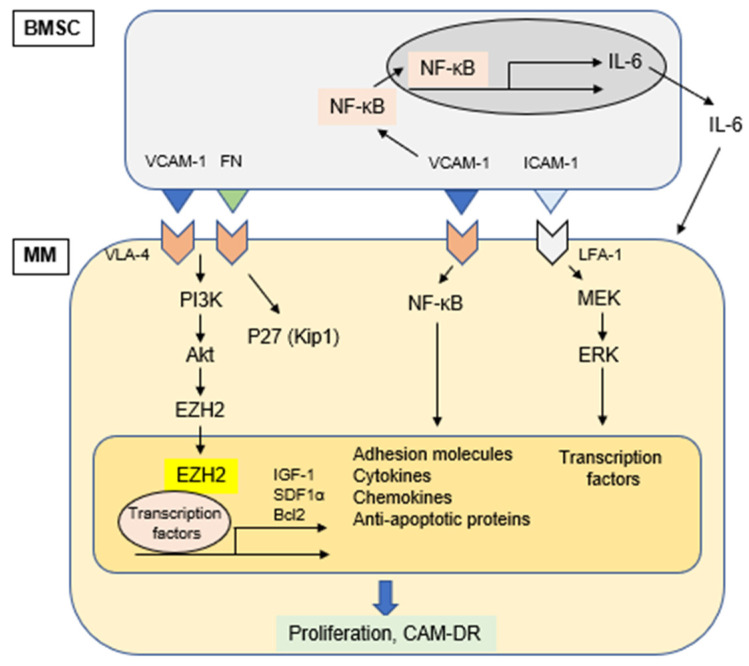
Cell adhesion-induced activation of signaling pathways.

**Table 1 cancers-13-00216-t001:** Representative soluble factors and activation of signaling pathways.

Cytokines	ERK	JAK-STAT	AKT	NF-κB
APRIL	–	–	–	✓
BAFF	–	–	–	✓
CD40-L	–	–	–	✓
FGF	✓	–	–	–
HB-EGF	–	–	✓	–
HGF	✓	✓	✓	–
IGF-1	–	–	✓	✓
IL-6	✓	✓	✓	–
IL-10	–	✓	–	–
OSM	✓	✓	✓	–
SDF-1α	✓	–	✓	✓
TLR-L	–	–	–	✓
TNFα	✓	–	–	✓
VEGF	✓	–	✓	–

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
