# Peer review of "Signaling Pathway Mediating Myeloma Cell Growth and Survival"

_cancers, 2021, doi:10.3390/cancers13020216_

Round 1

Reviewer 1 Report

The paper «Signaling Pathway Mediating Myeloma……” by Teru Hideshima and Kenneth C. Anderson is an update on a topic that has been under study for many years and where there is a lot of data to review. It is laudable that the authors try to sum up the most important knowledge in this rather complex field. They have themselves contributed substantially to this field over the years and it is understandable that the paper is influenced and somewhat biased by data from their own group.

The focus is on signaling pathways and molecules involved intracellularly and this is well covered in the manuscript. But equally important is the identity of exogenous molecules that are able to activate the pathways. A few years back, I made this list of soluble molecules that had been describes as myeloma growth factors: IL-3, IL-6, IL-10, IL-15, IL-21, OSM, GM-CSF, G-CSF, IGF-1, HB-EGF, VEGF, FGF, HGF, SDF-1α, MIP-1α, BAFF, APRIL, CD40-L, TLR-ligands and TNF. Since then, others factors have been added. The authors have chosen a selection of these molecules and added a few others, but do not try to justify their choices. The present paper appears as a list of interesting molecules and mechanisms, but without an attempt to rank them by importance.  Then, the only ranking is by what they include and what they omit. Some of the omissions are difficult to understand. HB-EGF and HGF are well documented myeloma cell growth factors and important angiogenic factors but not mentioned in the paper. In papers where the level of several growth factors have been measured in serum or bone marrow plasma and correlated to disease severity, HGF always comes out as strongly correlated. As an example, in a recent paper (doi: 10.1093/jncics/pkz104), 45 angiogenic factors were measured and EGF, HGF together with Ang-2 and SDF-1 came out as the only significant predictors for transition of MGUS to MM. The HGF signaling pathway also shows important crosstalk with IL-6 signaling (doi: 10.1111/j.1600-0609.2009.01212.x).

The paper would have been more interesting (and probably more provocative) if they had ventured to discuss the controversies and point at the “unknowns” in the field. Which of the growth factors are actually found in substantial amounts in MM bone marrow? Which of them are autocrine factors? Which of them correlate with disease severity? Which signaling pathways are aberrantly activated and which are just reflecting normal plasma cell signaling?

More specific points:

1) Line 39-41 and 80-82: Here the authors list cell types in the bone marrow. Why are bone marrow mononuclear cells included as if they were a specific cell type along with osteoblasts etc? A better approach wound have been to divide cells into hematopoietic cells and non-hematopoietic cells and to list the individual cell types under those headings. They mention BMSCs both places. On line 68 they introduce BM mesenchymal stromal cells. Are they different from BMSCs? Are BMSCs not mesenchymal?

2) Line 59-60: Here they claim that VLA-4 binds ICAM-1 without giving a reference. Correct? If so, they need to provide a reference. They also say that ICAM-1 and VCAM-1 on MM cells bind VLA-4 on BMSCs. In lines 400- 402 and in Fig 2 they say the opposite; VLA-4 on MM cells binds VCAM-1 on BMSCs. Are both correct?

3) B cell–activating factor is abbreviated BAF in lines 304 and 315, in other places with the more common acronym BAFF.

4) On line 127 they say that IL-3 is produced by MM cells. RNAseq data from the CoMMpass database suggest that this is not the case (doi: 10.1016/j.blre.2019.100646). Do they have a reference?

5) On line 133 the say: “mutation of either ERK, MEK or Raf has not yet been reported”. Several reports have found that a subset of patients carry an activating mutation (V600E) in the BRAF kinase (e.g.: doi: 10.1158/2159-8290.CD-13-0014).

6) There is a few typing errors or missing words throughout the manuscript, e.g. lines 135, 249, 306, 375 (references).

Author Response

Reviewer #1

Comments and Suggestions for Authors

The paper «Signaling Pathway Mediating Myeloma……” by Teru Hideshima and Kenneth C. Anderson is an update on a topic that has been under study for many years and where there is a lot of data to review. It is laudable that the authors try to sum up the most important knowledge in this rather complex field. They have themselves contributed substantially to this field over the years and it is understandable that the paper is influenced and somewhat biased by data from their own group.

The focus is on signaling pathways and molecules involved intracellularly and this is well covered in the manuscript. But equally important is the identity of exogenous molecules that are able to activate the pathways. A few years back, I made this list of soluble molecules that had been describes as myeloma growth factors: IL-3, IL-6, IL-10, IL-15, IL-21, OSM, GM-CSF, G-CSF, IGF-1, HB-EGF, VEGF, FGF, HGF, SDF-1α, MIP-1α, BAFF, APRIL, CD40-L, TLR-ligands and TNF. Since then, others factors have been added.

The authors have chosen a selection of these molecules and added a few others, but do not try to justify their choices. The present paper appears as a list of interesting molecules and mechanisms, but without an attempt to rank them by importance.  Then, the only ranking is by what they include and what they omit. Some of the omissions are difficult to understand. HB-EGF and HGF are well documented myeloma cell growth factors and important angiogenic factors but not mentioned in the paper. In papers where the level of several growth factors have been measured in serum or bone marrow plasma and correlated to disease severity, HGF always comes out as strongly correlated. As an example, in a recent paper (doi: 10.1093/jncics/pkz104), 45 angiogenic factors were measured and EGF, HGF together with Ang-2 and SDF-1 came out as the only significant predictors for transition of MGUS to MM. The HGF signaling pathway also shows important crosstalk with IL-6 signaling (doi: 10.1111/j.1600-0609.2009.01212.x).

We thank the Reviewer for this helpful comment. We newly generated new Table 1 that includes representative soluble factors and target signaling pathways. We also include a paragraph describing soluble factors that can impact MM disease progression and crosstalk between HGF and IL-6 in “Mediators for activation of MM signaling pathways” section (new lne56 – 69).

The paper would have been more interesting (and probably more provocative) if they had ventured to discuss the controversies and point at the “unknowns” in the field. Which of the growth factors are actually found in substantial amounts in MM bone marrow? Which of them are autocrine factors? Which of them correlate with disease severity? Which signaling pathways are aberrantly activated and which are just reflecting normal plasma cell signaling?

We thank the Reviewer for this suggestion and we include “Signal transduction in myeloma cells is very complicated in the BM microenvironment, and multiple signal transduction systems are simultaneously activated by multiple soluble factors. For example, one soluble factor may activate a plurality of signal transduction pathways at the same time, but a plurality of soluble factors may cooperate to induce more potent activation of the same signaling pathway. It should be also noted that the existence of other factors that could suppress myeloma cells growth cannot be ruled out. Therefore, it should be carefully evaluated whether the findings obtained in the preclinical studies actually occur in the BM environment of MM patients.” in “Conclusion” section.

More specific points:

1) Line 39-41 and 80-82: Here the authors list cell types in the bone marrow. Why are bone marrow mononuclear cells included as if they were a specific cell type along with osteoblasts etc? A better approach wound have been to divide cells into hematopoietic cells and non-hematopoietic cells and to list the individual cell types under those headings. They mention BMSCs both places. On line 68 they introduce BM mesenchymal stromal cells. Are they different from BMSCs? Are BMSCs not mesenchymal?

We agree with Reviewer’s comments and deleted “mononuclear cells” from line 40. To avoid confusion, we also deleted “of cells including BMSCs, osteoclasts, osteoblasts, vascular endothelial cells, and dendritic cells” from line 80-81.

2) Line 59-60: Here they claim that VLA-4 binds ICAM-1 without giving a reference. Correct? If so, they need to provide a reference. They also say that ICAM-1 and VCAM-1 on MM cells bind VLA-4 on BMSCs. In lines 400- 402 and in Fig 2 they say the opposite; VLA-4 on MM cells binds VCAM-1 on BMSCs. Are both correct?

We thank the Reviewer for this comment and apologize for confusion.  We corrected “intercellular adhesion molecule 1 (ICAM-1) and vascular cell adhesion molecule 1 (VCAM-1) on MM cells bind very late antigen (VLA-4) on BMSCs” to “intercellular adhesion molecule 1 (ICAM-1) and vascular cell adhesion molecule 1 (VCAM-1) on MM cells bind lymphocyte function-associated antigen-1 (LFA-1) and very late antigen (VLA-4) on BMSCs, respectively”. We also included references accordingly.

To avoid repeating same sentence, we deleted “For example, intercellular adhesion molecule 1 (ICAM-1) and vascular cell adhesion molecule 1 (VCAM-1) on BMSCs cells bind to very late antigen (VLA-4) on MM cells” (Line 400-402).

3) B cell–activating factor is abbreviated BAF in lines 304 and 315, in other places with the more common acronym BAFF.

We corrected “BAF” to “BAFF”.

4) On line 127 they say that IL-3 is produced by MM cells. RNAseq data from the CoMMpass database suggest that this is not the case (doi: 10.1016/j.blre.2019.100646). Do they have a reference?

We included reference “IL-3 expression by myeloma cells increases both osteoclast formation and growth of myeloma cells” (Lee JW et al., Blood 2004) in the text. We also modified the sentence.

5) On line 133 the say: “mutation of either ERK, MEK or Raf has not yet been reported”. Several reports have found that a subset of patients carry an activating mutation (V600E) in the BRAF kinase (e.g.: doi: 10.1158/2159-8290.CD-13-0014).

We agree with Reviewer’s comment and deleted “Raf” from the sentence (new line147).

6) There is a few typing errors or missing words throughout the manuscript, e.g. lines 135, 249, 306, 375 (references).

We corrected them accordingly.

Reviewer 2 Report

The review articel entitled „Signaling Pathways Mediating Myeloma Cell Growth and Survival“ written by Teru Hideshima and Kenneth C. Anderson gives a decent overview about the different signaling pathways that mediate muliple myeoma (MM) growth and survival.

Specifically, the authors highlight a very important topic in MM, namely the crosstalk or physical interaction of MM cells with their microenvironment that consists of different cells such as mesenchymal bone marrow stroma cells, osteoblasts, osteoclasts endothelial and dendritic cells. Due to the crosstalk which can be mediated via soluble factors such as cytokines and growth factors, but also via  cell-cell adhesion, MM cells as well as the different cells surrounding the tumor express and secrete proteins such as IL6 that on the one hand induce  MM cell growth and on the other hand promote osteoclastogenesis and angiogenesis.

The three main routest to activate survival pathways in MM is via 1. soluble factors 2.  adhesion 3. exosomes.

Minor comments:

1.

The first two routes of activation were nicely addressed in the review, however the third route via exosomes was only mentioned in the first section, but was no longer addressed in the following.

The reviewer suggests that the third route via exosomes is handled like the route via soluble factors and adhesion molecules. If this is not possible due to e.g. lack of space, the reviewer suggests to make a note in section 3.

2.

Line 80 to 94 seem to be redundant in major parts

3.

Lane 89: The cytokines…

To which molecules of the previous sentence do the authors refer? The authors mention several growth factors in the previous sentence.

4.

Lane 113-114

What kind of interaction? Via soluble factors, via adhesion? Which factors are involved?

5.

Lane 113-128:  connectives are either missing or misleading for a proper understanding

6.

Section 5

It would be helpful for the reader to shortly mention in the beginning of each sub-section which cells are invovled, e.g. which factors expressed by which cells lead to the activation of the MEK-ERK , JAK/STAT, PI3K-signaling etc..

Sometimes this information is also missing within the text. Sine this is a basic feature of the review, it is important that the reader can follow which molecule is expressed by which cell and stimulates which cell. This is getting specifically important when the same molecule is expressed by different cells with different functions.

7.

Lane 133:

Mutations in BRAF (~4% of primary MM), CRAF and ARAF are published

8.

Lane 136:

Tumor mutational burden (TMB)?

9.

Lane 139: citation is missing

10.

Lane 141: there is also literature saying that NRAS and KRAS mutaitons are present in MGUS. Could the authors please check this issue again?

11.

Lane 169: literature is missing, connecting passage to RAF inhibitors?

12.

Lane168-182: The authors switch from MEK inhibitors to RAF inhibitors and back to MEK inhibitors which is confusing.

13.

Lane 185: The reviewer does not understand the use of the word however

14.

Lane 210: it has been shown that miRNA21 which was found tob e regulated by STAT3, also promotes…..

To mention only miRNA21 without the connection to STAT3 is confusing at that point.

15.

Lane 257: According to the sentence in lane 56 one expects a listing of only protein kinases. It might be better to delete the word protein kinases in lane 256 and then mention it in lane 257 before GSK?

16.

Lane 265: literature is missing, RAS-MEK-ERK

17.

267-301

It would be easier if the authors would first explain the role of the molecules, then make a general statement about the possible use of inhibitors and then mention the different examples.

This applies to all sections.

Lane 287: reference is missing

The authors often say „We have also“. Sometimes this fits, but sometimes it does not seem suitable fort he reviewer. E.g. in lane 294. One gets the impression that all the investigations with MK2206 were also done by the authors. If this is the case, this should be made more clear before.  Otherwise, one may use another phrase.

18.

Lane 304: … in MM cells upon contact with cells from the BM ….

19.

Lane 349: …to block the non-canonical…

20.

Lane 365: literature missing

However, it is known thatthe canonical WNT…

21.

Lane 367: literature missing

22.

Lane 370: literature missing

23.

Lane 400: the reviewer does not understand this sentence. Could the authors please be more precise?

24.

Lane 404: only via NFkB?

25.

Lane 406: At that point the authors could maybe remeber the reader that  IL6 is a soluble factor that regulates growth in MM cells via the JAK/STAT signaling pathway but is also involved in osteoclastogenesis and angiogenesis

The authors may cite also newer literature in this context (e.g. Harmer et al., Frontiers in Endocrinology, 2019)

26.

Lane 426: As describes in section 5.4, bortezomib….

27.

Lane 432-434; The abbreviation PBGF is confusing. In NCBI the alias is PBSF or SDF1. It is also not getting clear from the text that SDF1 is the alias fro CXCL12. The reviewer suggests to introduce CXCL12 saying directly that it is also called SDF1, follwed by an explanation of this molecule.

28.

Lane 441-442; …murine MM-models, suggesting an epigenetic mecahnism….

29.

Lane 446-448: Would the authors say that a combination with different drugs is absolutely neccessary?

….may represent promising novel therapeutic strategies…?

Author Response

Reviewer #2

Comments and Suggestions for Authors

The review articel entitled „Signaling Pathways Mediating Myeloma Cell Growth and Survival“ written by Teru Hideshima and Kenneth C. Anderson gives a decent overview about the different signaling pathways that mediate muliple myeoma (MM) growth and survival.

Specifically, the authors highlight a very important topic in MM, namely the crosstalk or physical interaction of MM cells with their microenvironment that consists of different cells such as mesenchymal bone marrow stroma cells, osteoblasts, osteoclasts endothelial and dendritic cells. Due to the crosstalk which can be mediated via soluble factors such as cytokines and growth factors, but also via  cell-cell adhesion, MM cells as well as the different cells surrounding the tumor express and secrete proteins such as IL6 that on the one hand induce  MM cell growth and on the other hand promote osteoclastogenesis and angiogenesis. 

The three main routest to activate survival pathways in MM is via 1. soluble factors 2.  adhesion 3. exosomes.

Minor comments:

  1. The first two routes of activation were nicely addressed in the review, however the third route via exosomes was only mentioned in the first section, but was no longer addressed in the following.

The reviewer suggests that the third route via exosomes is handled like the route via soluble factors and adhesion molecules. If this is not possible due to e.g. lack of space, the reviewer suggests to make a note in section 3.

As suggested, we added “Although exosomes play an important role on MM cell pathogenesis in the BM microenvironment, they cannot be classified as classical soluble factors and are not further discussed” in Section3 (new Line 91-93).

  1. Line 80 to 94 seem to be redundant in major parts

As suggested, we shorten the sentences as “The BM microenvironment consists of different cell types which produce and secrete numerous soluble factors and physically interact with MM cells. Importantly, soluble activating factors are further upregulated by MM cell adhesion with these cells. These cytokines activated signaling pathways include: extracellular signal related kinase (ERK, also known as p42/44 mitogen activated protein kinase); Janus kinase (JAK)2/signal transducers and activators of transcription (STAT)3; phosphoinositide 3-kinases (PI3K)/Akt (also known as protein kinase B); nuclear factor κ-B (NF-κB); transforming growth factor (TGF)-β/Smad and Wnt/β-catenin”.

  1. Lane 89: The cytokines…

To which molecules of the previous sentence do the authors refer? The authors mention several growth factors in the previous sentence.

The sentence has been deleted.

  1. Lane 113-114: What kind of interaction? Via soluble factors, via adhesion? Which factors are involved?

As suggested, we wrote new paragraph (new line 123-127).

  1. Lane 113-128:  connectives are either missing or misleading for a proper understanding

We thank for Reviewer’s comment. We deleted bone absorption part from this paragraph, since it is not relevant for this review.

  1. Section 5: It would be helpful for the reader to shortly mention in the beginning of each sub-section which cells are invovled, e.g. which factors expressed by which cells lead to the activation of the MEK-ERK , JAK/STAT, PI3K-signaling etc..

Sometimes this information is also missing within the text. Sine this is a basic feature of the review, it is important that the reader can follow which molecule is expressed by which cell and stimulates which cell. This is getting specifically important when the same molecule is expressed by different cells with different functions.

We thank the Reviewer for this comment. We think most representative cytokines secreted by different cell types are already demonstrated in Figure 1. As requested, we generated new Table 1 to summarize activation of signaling pathways induced by each representative cytokine.

  1. Lane 133: Mutations in BRAF (~4% of primary MM), CRAF and ARAF are published

We agree with the reviewer.  Please see reply to comment #5 by Reviewer #1.

  1. Lane 136: Tumor mutational burden (TMB)?

It is indeed “tumor burden”.

  1. Lane 139: citation is missing

We inserted a reference, as suggested.

  1. Lane 141: there is also literature saying that NRAS and KRAS mutaitons are present in MGUS. Could the authors please check this issue again?

It is correct statement from the reference. We added “Spanish population” at the end of sentence.

  1. Lane 169: literature is missing, connecting passage to RAF inhibitors?

We included a reference accordingly.

We also wrote following paragraph; “Whole-genome sequencing data have recently revealed that a subset of patients possesses an activating mutation (V600E) in the BRAF kinase. The BRAF V600E mutation triggers constitutive activation of Ras-Raf-MEK-ERK signaling pathway, stimulating cellular growth, differentiation and survival. However, it is still unclear whether constitutively activated MEK-ERK pathway in the cells with BRAF V600E is further enhanced by soluble activating factors in the BM microenvironment.  Previous study showed that BRAF V600E was detected in 2.8% of patients with symptomatic myeloma and 1.8% of all patients with monoclonal plasma cell disorders, respectively. Another study also showed that BRAF V600E was detected in 5.3% of the patients with MM. Importantly, patients with BRAF V600E showed significantly higher incidence of extramedullary disease and a shorter overall survival. Sorafenib is an orally available compound which predominantly inhibits Raf kinase and VEGF receptor 2. In a phase II clinical trial (SWOG S0434) of sorafenib as monotherapy in relapsed and refractory MM (RRMM), it did not exhibit clinical activity. Further research should focus on combination therapy of sorafenib with standard treatments in selected patients with RR MM. For example, sorafenib upregulates Akt phosphorylation, and its use in combination with rapamycin to inhibit mTOR showed synergistic growth inhibitory effects”.

  1. Lane168-182: The authors switch from MEK inhibitors to RAF inhibitors and back to MEK inhibitors which is confusing.

As suggested, we moved “Raf inhibition” to next (new) paragraph. Please see above.

  1. Lane 185: The reviewer does not understand the use of the word however

 We replaced “However” with “Importantly”.

  1. Lane 210: it has been shown that miRNA21 which was found tob e regulated by STAT3, also promotes…..

To mention only miRNA21 without the connection to STAT3 is confusing at that point.

We agree with Reviewer’s comment. miRNA-induced STAT3 signaling is not relevant in this review and we deleted sentences of miRNA from the paragraph.

  1. Lane 257: According to the sentence in lane 56 one expects a listing of only protein kinases. It might be better to delete the word protein kinases in lane 256 and then mention it in lane 257 before GSK?

As suggested, we modified sentences accordingly.

  1. Lane 265: literature is missing, RAS-MEK-ERK

As, suggested, we added references.

  1. 267-301

It would be easier if the authors would first explain the role of the molecules, then make a general statement about the possible use of inhibitors and then mention the different examples. 

This applies to all sections.

We do understand the comment of the Reviewer. We describe some representative inhibitors to target specific signaling pathways; however, the main purpose of this review is to describe how differential signaling pathways are activated in MM cell in the context of the bone marrow microenvironment, with potential clinical implications, rather than mentioning specific inhibitors. 

Lane 287: reference is missing

As, suggested, we added reference.

The authors often say „We have also“. Sometimes this fits, but sometimes it does not seem suitable fort he reviewer. E.g. in lane 294. One gets the impression that all the investigations with MK2206 were also done by the authors. If this is the case, this should be made more clear before.  Otherwise, one may use another phrase.

As suggested, we deleted “also” from the sentence.

  1. Lane 304: … in MM cells upon contact with cells from the BM ….

We think it is fine as described. since NF-κB can be activated by variety of soluble factors in the BM microenvironment.

  1. Lane 349: …to block the non-canonical…

We think it is fine as described. Proteasome inhibitors can block both canonical and non-canonical NF-κB pathways.

  1. Lane 365: literature missing

 However, it is known that the canonical WNT…

We corrected the sentence to “The biologic significance of Wnt-β-catenin signaling in MM has been studied” with references.

  1. Lane 367: literature missing

We added reference, accordingly.

  1. Lane 370: literature missing

We added reference, accordingly.

  1. Lane 400: the reviewer does not understand this sentence. Could the authors please be more precise?

We modified the sentence as “This cell-cell adhesion is mediated via matching of specific adhesion molecules on each cell surface; VLA-4 on MM cells and VCAM-1 on BMSC”.

  1. Lane 404: only via NFkB?

The sentence does not mean that upregulation of production of cytokines and the expression of adhesion molecules are mediated solely by NF-κB activity. However, it is well known that activation of NF-κB upregulates certain cytokines (eg., IL-6) and adhesion molecules (eg., VCAM-1).

  1. Lane 406: At that point the authors could maybe remeber the reader that  IL6 is a soluble factor that regulates growth in MM cells via the JAK/STAT signaling pathway but is also involved in osteoclastogenesis and angiogenesis 

The authors may cite also newer literature in this context (e.g. Harmer et al., Frontiers in Endocrinology, 2019)

As suggested, we added sentence of “Of note, IL-6 also enhances angiogenesis via VEGF stimulation and modulate osteoclast differentiation (Harmer D., et al. 2019, Frontiers in Endocrinology) (new line 414-415).

  1. Lane 426: As describes in section 5.4, bortezomib….

The sentence was corrected as suggested.

  1. Lane 432-434; The abbreviation PBGF is confusing. In NCBI the alias is PBSF or SDF1. It is also not getting clear from the text that SDF1 is the alias fro CXCL12. The reviewer suggests to introduce CXCL12 saying directly that it is also called SDF1, follwed by an explanation of this molecule.

We corrected the sentence, as suggested.

  1. Lane 441-442; …murine MM-models, suggesting an epigenetic mecahnism….

We corrected the sentence, as suggested.

  1. Lane 446-448: Would the authors say that a combination with different drugs is absolutely neccessary? 

….may represent promising novel therapeutic strategies…?

We agree with Reviewer’s comment and corrected the sentence accordingly.